# Multi-compartmental diversification of neutralizing antibody lineages dissected in SARS-CoV-2 spike-immunized macaques

Marco Mandolesi [1] ✉, Hrishikesh Das [2], Liset de Vries [1], Yiqiu Yang[1], Changil Kim[1], Manoj Dhinakaran [1], Xaquin Castro Dopico [1], Julian Fischbach [1], Sungyong Kim[1], Mariia V. Guryleva[1], Monika Àdori[1], Mark Chernyshev [1], Aron Stålmarck[1], Leo Hanke [1], Gerald M. McInerney [1], Daniel J. Sheward [1], Martin Corcoran[1], B. Martin Hällberg [2], Ben Murrell [1,3] & Gunilla B. Karlsson Hedestam [1,3] ✉

The continued evolution of SARS-CoV-2 underscores the need to understand qualitative aspects of the humoral immune response elicited by spike immunization. Here, we combine monoclonal antibody (mAb) isolation with deep B cell receptor (BCR) repertoire sequencing of rhesus macaques immunized with prefusion-stabilized spike glycoprotein. Longitudinal tracing of spike-sorted B cell lineages in multiple immune compartments demonstrates increasing somatic hypermutation and broad dissemination of vaccine-elicited B cells in draining and non-draining lymphoid compartments, including the bone marrow, spleen and, most notably, periaortic lymph nodes. Phylogenetic analysis of spike-specific monoclonal antibody lineages identified through deep repertoire sequencing delineates extensive intra-clonal diversification that shaped neutralizing activity. Structural analysis of the spike in complex with a broadly neutralizing mAb provides a molecular basis for the observed differences in neutralization breadth between clonally related antibodies. Our findings highlight that immunization leads to extensive intra-clonal B cell evolution where members of the same lineage can both retain the original epitope specificity and evolve to recognize additional spike variants not previously encountered.

Humoral immune responses against viral spike glycoproteins stimulated by infection or vaccination are characterized by polyclonal repertoires of antibodies that target distinct spike epitopes. Of these, a proportion mediates virus neutralization, and blocking viral entry into target cells. Viruses that cause chronic infections such as HIV-1, or globally persistent viruses such as Influenza virus, have evolved immune evasion strategies to circumvent host antibody responses, ensuring their continued transmission. Such escape mechanisms include sequence variations that abolish epitope recognition, as well as glycan and conformational shielding that limit access to functionally conserved epitopes targeted by broadly neutralizing antibodies[1].

During the SARS-CoV-2 pandemic, antibody escape mutations have increasingly evolved in the spike glycoprotein, curtailing population immunity established by previous infections and vaccinations[2–6]. All variants known to be currently circulating in humans are descendants of Omicron, carrying multiple immune escape mutations whose

[1]Department of Microbiology, Tumor and Cell Biology, Karolinska Institutet, Stockholm, Sweden. [2]Department of Cell and Molecular Biology, Karolinska Institutet, Stockholm, Sweden. [3]These authors contributed equally: Ben Murrell, Gunilla B. Karlsson Hedestam. ✉e-mail: marco.mandolesi.job@gmail.com; gunilla.karlsson.hedestam@ki.se

molecular basis is typically well understood[7–9]. Many such mutations were independently acquired in different sub-lineages through convergent evolution[10].

Following repeated or prolonged exposure to antigen, the host antibody response evolves via B cell diversification through somatic hypermutation (SHM) in germinal center (GC) reactions[11,12]. Studies have shown that SHM-mediated affinity maturation of SARS-CoV-2 neutralizing antibodies can to some degree overcome immune escape[13,14]. How antigen-specific B cell lineages evolve following prime-boost immunizations with the ancestral D614G SARS-CoV-2 spike (lineage B.1), and how this influences the development of neutralization breadth, is of interest as most vaccinated people worldwide received the first-generation vaccines based on the ancestral D614G spike.

To date, most studies aimed at characterizing SARS-CoV-2 humoral responses have relied on the analysis of polyclonal plasma, providing information about the overall activity of circulating antibodies, and the isolation of spike-specific monoclonal antibodies (mAbs), allowing definition of genetic and functional properties of individual specificities. While these approaches are highly informative, they do not provide in-depth information about B cell lineage development following antigen exposure. A powerful approach to address this question is to combine mAb isolation with bulk B cell receptor (BCR) repertoire sequencing to identify clonally related sequences of antibodies with known specificities in rich repertoire datasets, allowing in-depth interrogation of B cell lineage evolution[15–19].

Determining the distribution of B cell lineages across diverse tissues is also of great interest to understand the anatomical dissemination of B cell responses, which may affect disease protection. While human studies are often limited to the analysis of blood samples, the macaque model offers opportunities to characterize B cell lineages across multiple immune compartments, including draining lymph nodes and non-draining lymphoid tissues, such as the spleen and bone marrow (BM)[16]. The BM is of particular interest as the site of long-lived plasma cells. While it was recently shown that the BM harbors spike-specific plasma cells induced by both SARS-CoV-2 infection and by spike vaccination[20–22], our knowledge about B cell lineage evolution and anatomical distribution of spike-specific B cell lineages in the BM and other immune tissues remains limited.

Here, we performed in-depth analyses of rhesus macaques immunized four times with recombinant pre-fusion stabilized spike in Matrix-M adjuvant. We isolated immunoglobulin (IG) heavy and light chain (HC and LC) sequences from a large set of spike-sorted B cells. From a subset of these, we cloned HC-LC pairs for mAb production to identify neutralizing antibodies. We then used the HC sequences to query deep BCR (IgG) repertoire data from longitudinally sampled animals to assess the level of expansion and anatomical distribution of antigen-specific B cells. Our findings demonstrate extensive dissemination of spike-sorted B cell lineages in BM, spleen, and non-draining lymph nodes. A particularly high number of queried lineages were traced to the periaortic lymph nodes (perLN), suggesting that this is a central site for B cell circulation. We further characterized the evolution of selected neutralizing antibody lineages, including a broadly neutralizing lineage based on mAb23, for which SHM-induced diversification resulted in members that displayed either more focused neutralizing activity or increased breadth against Omicron sub-variants, providing a deeper understanding of Ab repertoire development following vaccination.

## Results

### Rhesus macaque spike-sorted B cells use a broad range of IG V genes

We used pre-fusion stabilized ancestral D614G spike trimers, adjuvanted in Matrix-M, to immunize rhesus macaques (H03 and I10) at weeks 0, 4 and 9, as previously reported[23]. Peripheral blood mononuclear cells (PBMCs), BM and draining inguinal lymph nodes (iLN) were sampled throughout the immunization regimen. A final boost of I10 and H03 was performed in week 30 or 31, respectively. For H03, samples from BM, spleen, iLN, mesenteric LN (mesLN), axillary LN (axLN), and periaortic LN (perLN) were obtained at termination, one week after the last boost (week 32) (Fig. 1a and Supplementary Fig. 1A). For I10, blood samples were collected at week 32, with additional blood, iLN, spleen and BM samples obtained at termination in week 46. Using a fluorescently labeled trimeric spike probe on samples collected at termination, we used FACS to isolate B cells from two distinct iLN (iLN-R1 and iLN-R2), as well as from spleen and a pool of non-draining LNs (mesLN, axLN, and perLN) for single-cell V(D)J sequencing using Chromium (10X Genomics) (Supplementary Fig. 1B). The same process was performed on PBMCs obtained at week 32 for I10. Cell hashing barcodes were used to label sequences according to their compartment of origin. Bulk HC sequences were obtained from longitudinal samples via deep repertoire sequencing. B cell lineage tracing was performed using both the single cell and the bulk HC sequences. For accurate lineage identification and SHM characterization, we genotyped the animals for their HC and LC variable (V) and junctional (J) germline gene/allele content using IgDiscover[24] (Supplementary Fig. 1C, D and Supplementary Data 1).

Single-cell sequence analysis of spike-sorted B cells identified a total of 1250 paired HC and LC sequences, of which 1048 were unique (Supplementary Data 2). Analysis of IG heavy chain V (IGHV) gene usage demonstrated the utilization of a wide range of genes. Most of these were also frequently used in the total IgG repertoire of the animals (Fig. 1b). The IGHV gene frequency in the total IgG repertoire was consistent with that observed in 15 other rhesus macaques reported in Chernyshev et al.[25]. When analyzing IG kappa and lambda chain V (IGKV and IGLV) gene usage in the single-cell data, we found that the LCs were used in combination with a wide variety of HCs, with no dominating HC-LC pairs observed (Fig. 1c).

### Accumulation of SHM, extensive compartmental dissemination, and persistence of early elicited spike-sorted B cell lineages

For each animal, we used the spike-sorted sequences from samples obtained at termination as queries to trace the respective lineages in longitudinal samples collected throughout the immunization regimen. A total of 191 and 41 lineages were traced in H03 and I10, respectively (Supplementary Data 2). A notable finding was that queried Ab lineages identified early were recurrently traced at subsequent time points following immunization (Fig. 2). In H03, 85.7% of the queried lineages detected in blood 2 weeks after priming were identified throughout the samples obtained from later timepoints, demonstrating that these lineages were persistent.

Longitudinal examination of the traced sequences in both H03 and I10 demonstrated increasing intra-lineage accumulation of SHM after each immunization (Fig. 2 and Supplementary Fig. 2A). In H03, almost all queried lineages (92.2%) were traced in at least one of the NGS libraries from samples obtained at termination (week 32). Of these, a very high proportion (89%) were traced in the perLN, suggesting that this is an active site for B cell recirculation (Fig. 2). Many queried lineages were also traced in BM samples (20.4% at week 6, 24.1% at week 11, and 58.6% at week 32) collected after the second, third, and fourth immunizations respectively, several of which overlapped (Fig. 2 and Supplementary Fig. 2B). Fewer queried lineages were traced in blood in the period between immunization 3 and 4 (5.7% at week 19, 5.2% at week 23 and 3.7% at week 27; Fig. 2), during which time the plasma neutralizing Ab titers decreased[23]. Queried lineages steadily accumulated SHM irrespective of the compartment they were traced in, suggesting dissemination of affinity-matured B cells throughout the body (Fig. 2 and Supplementary Fig. 2C).

To identify neutralizing Abs, we generated a set of mAbs from H03. Out of a total of 38 spike-binding mAbs, 13 were directed against the spike receptor binding domain (RBD) and 10 displayed potent

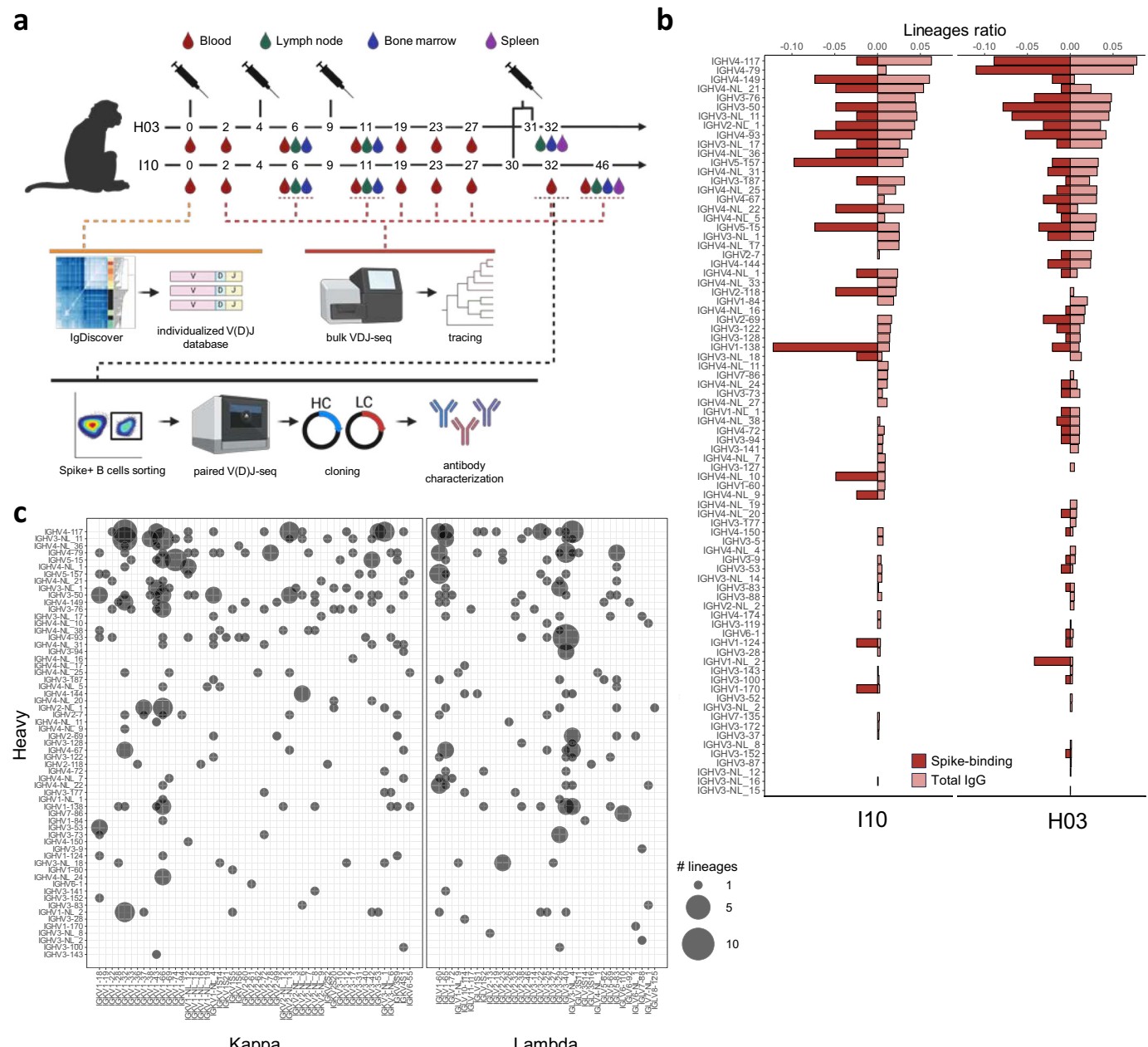

**Fig. 1 | Study design and immunoglobulin gene usage of vaccine-induced spike-binding B cells. a** Study overview. Two Chinese-origin rhesus macaques, H03 and I10, were inoculated four times with a pre-fusion stabilized spike ectodomain adjuvanted with Matrix-M at weeks 0, 4, 9. Additional boosting was performed at week 31 for H03 and week 30 for I10. Samples (blood, LNs, BM and spleen) were collected as indicated and used to produce bulk IgG HC libraries. Samples from week 0 were used for IgM, IgK and IgL library production and immunoglobulin genotyping with IgDiscover. Samples from week 32 were used to sort spike-binding B cells, which were processed for 10X single-cell V(D)J sequencing and mAb cloning. Figure created with BioRender.com. **b** Comparison of *IGHV* usage in the total IgG repertoire (light red) and the spike-sorted repertoire (dark red). The y-axis shows *IGHV* genes ranked in descending order based on frequency usage in the bulk IgG repertoires. Frequencies are displayed on the x-axis and calculated based on the number of total IgG (*n* = 48,528 and *n* = 51,904 for H03 and I10, respectively) vs spike-sorted (*n* = 191 and *n* = 41 for H03 and I10, respectively) for each *IGHV* gene. Source data are provided as a Source Data file. **c** Combined *IGHV* (y-axis), *IGKV* (x-axis left) and *IGLV* (x-axis right) genes identified in single-cell sequence data from both animals. Each combination is represented as a circle with the size directly proportional to the number of lineages computed with each *IGHV* and *IGKV* or *IGLV* combination. Source data are provided as a Source Data file.

neutralizing activity (Fig. 3a, b and Supplementary Data 3). Among the 38 mAbs, 36 belonged to 30 lineages, which were readily traced in IgG repertoires from at least one compartment (Fig. 3c), while the remaining 2 mAbs could not be traced. Consistent with the observation of H03 queried lineages, we found that IgG repertoires generated from samples collected at termination, after the fourth immunization, were particularly rich for spike-binding lineages. At this time point, all 30 lineages were traced in perLN, and most lineages were also detected

in spleen (83.3%), axLN (83.3%), and BM (76.6%) from week 32. A high proportion of lineages were also detected in blood (73.3%) and iLN (60.0%) 2 weeks after the second immunization at the week 6 time point. Five of the lineages were detectable in blood as early as week 2, after only one immunization, while all but two lineages were traced at week 6 in either blood, LN, or BM (Fig. 3c). The tracing results for all spike-binding lineages for which representative mAbs were not cloned are shown in Supplementary Fig. 3.

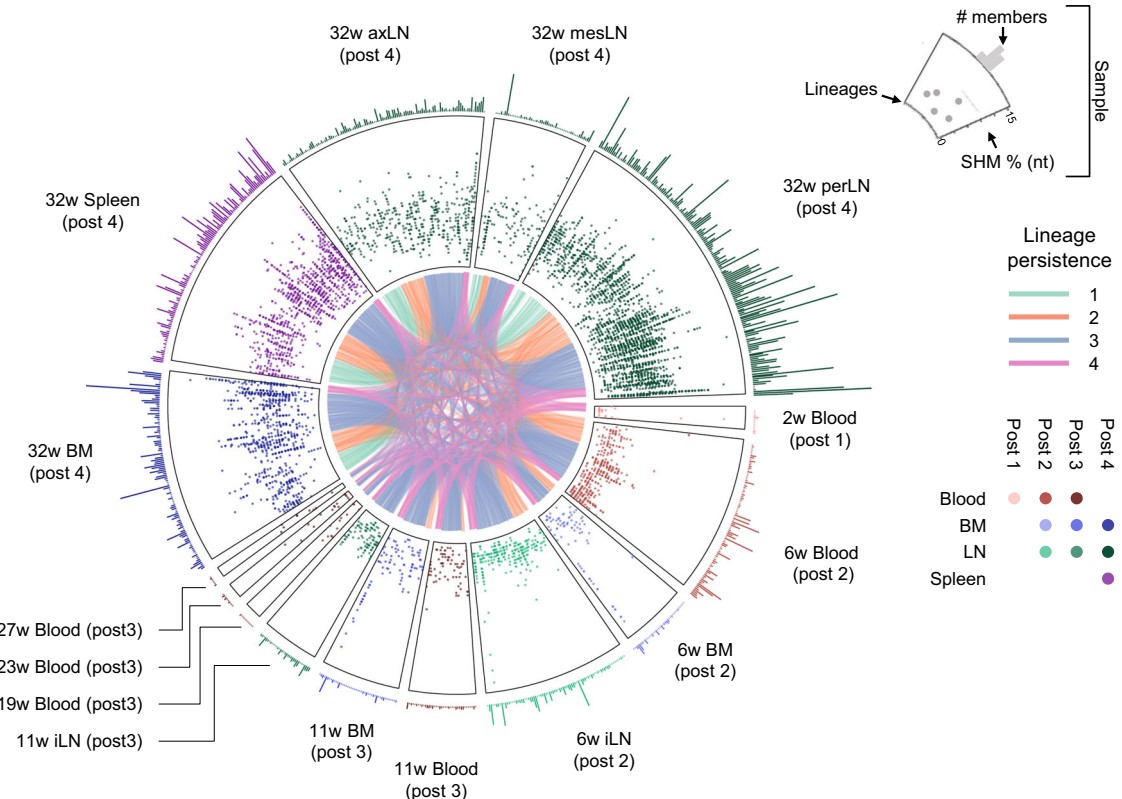

**Fig. 2 | Multicompartmental longitudinal analysis of spike-elicited B cell lineage evolution.** Each sector represents a sample from H03. Starting from degree 0, the samples are ordered clockwise by sample time point. The base of each sector represents a curved x-axis with each lineage as a point. Each sector is divided into three levels. The outer level shows the number of members of that lineage, the mid-level shows each member's SHM, and the inner common level connects lineages identified in multiple samples. The inner level is color-coded based on lineage persistence as determined by their identification at multiple time points (post 1, post 2, post 3, and post 4) in the traced data. The mid and outer sectors are color-coded with red, blue, green, and purple for blood, BM, LN, and spleen, respectively, with darker shades for increasing weeks. Source data are provided as a Source Data file.

When the genetic and functional properties of the 10 neutralizing mAbs were examined, two selected mAbs were clonally related; defined as using the same HC and LC V and J alleles and having similar HC and LC CDR3s (Fig. 3d). The SHM levels of the neutralizing mAbs at the at the nucleotide level varied between 2.6% and 6.1% for the HC and 1.1% and 9.5% for the LC, while at the amino acid level varied between 5.0% and 12.2% for the HC, and 3.2% and 17.4% for the LC. As many as 9 of the 10 neutralizing mAbs displayed highly potent activity against the vaccine strain D614G, with $IC_{50}$ titers below 0.01 mg/ml (Fig. 3d).

### Deep BCR repertoire tracing of spike-specific mAbs identified highly diversified lineages

We next focused our analysis on a selection of five potent neutralizing mAbs, the clonally related mAbs 21 and 51, as well as mAb 7, mAb 23, and mAb 44. We first examined their neutralization breadth against Beta and a panel of Omicron subvariants, BA.1, BA.2, BA.2.75, and BA.5. All mAbs demonstrated neutralizing activity against at least one variant beyond the ancestral D614G, which was used as an immunogen. (Fig. 4a). Three mAbs: mAb 21, mAb 23, and mAb 44, potently neutralized several Omicron variants, with mAb 23 displaying the broadest activity, neutralizing all seven tested variants.

To characterize the evolutionary pathways of B cells encoding these five neutralizing mAbs, we studied lineages 25287 (mAb 21, mAb 51), 10644 (mAb 23), 2319 (mAb 44), and 21180 (mAb 7). We used all somatic variants identified in the IgG repertoire tracing data to infer antibody lineage phylogenies (Fig. 4b). Each Ab lineage displayed unique maturation trajectories, with accumulation of SHM over time observed in all lineages. IgG repertoire sequencing data from the perLN compartment was prevalent across these phylogenies,

reflecting its global prevalence among traced spike-targeting lineages. For all four lineages, we also found variants identified in the blood, LN, spleen, and BM IgG repertoires distributed throughout the tree. Occasionally, we observed sub-branches comprising members from different time points that shared the same evolutionary distance from the inferred germline sequence, such as lineage 10644 and 21180, indicating early dissemination of the B cell lineage across different immune compartments. Examination of phylogenetic trees of the other five neutralizing mAb lineages, 34709 (mAb 13), 17512 (mAb 14), 50667 (mAb 42), 27358 (mAb 55), and 24725 (mAb 6) yielded similar results (Supplementary Fig. 4). Collectively, these results demonstrate the widespread intra-clonal dissemination and diversification of individual B cell lineages following vaccination.

### The mAb 23 lineage contains members with distinct neutralization breadth

To further examine B cell evolution influencing neutralization capability, we focused on lineage 10644. This lineage comprises mAb 23, which was cloned from the iLN-R2 compartment and identified as the most broadly neutralizing mAb in this study. Figure 5a displays the phylogenetic tree of this lineage and summarizes the time point and compartment from which each member of the lineage was identified, the amino acid sequences of each member compared to the germline *IGHV3-50*01* sequence, and the neutralizing activities of additional mAbs that were cloned and analyzed for their neutralizing activity.

The earliest time point at which we could trace this lineage was in IgG repertoire data from blood and iLN at week 6. We also found clonally related sequences at week 11 in blood and iLN, and at week 32 in BM, spleen, axLN and perLN. Interestingly, we identified the

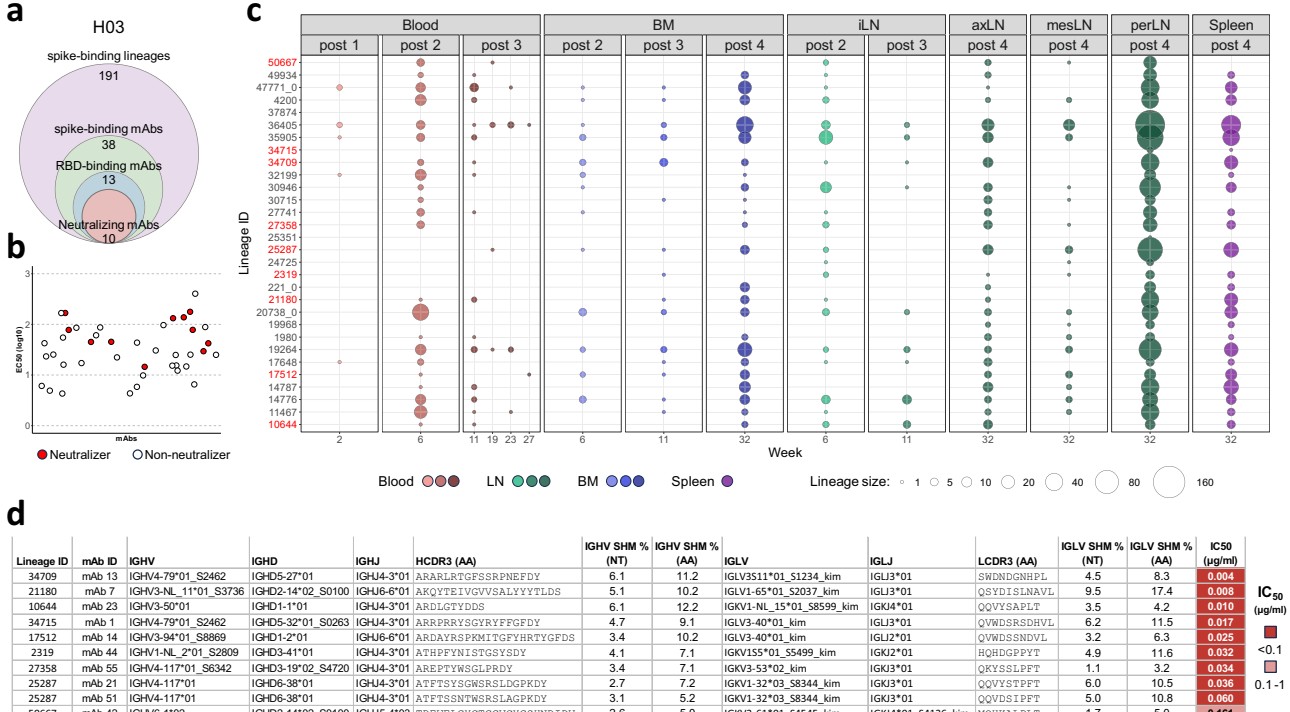

**Fig. 3 | Functional analysis of spike-binding mAbs and their lineage tracing in bulk IgG repertoires. a** Summary of the number of spike-binding lineages and cloned spike-binding, RBD-binding and neutralizing mAbs isolated from H03. **b** ELISA EC$_{50}$ values based on binding to the ancestral prefusion D614G spike protein with neutralizing mAbs highlighted in red and non-neutralizing mAbs in white. Binding assays were performed in triplicates for each mAb. Source data are provided as a Source Data file. **c** Dissemination of spike-binding mAb lineages where the bubble size is directly proportional to the number of sequences identified for each lineage in each library. The x-axis of each section shows the sample compartment, grouped by time point with the sampling week indicated. Source data are provided as a Source Data file. **d** Summary table displaying V, D and J allele assignments, V gene SHM levels, CDR3 amino acid identity and IC$_{50}$ neutralizing titers against the D614G variant for each neutralizing mAb. Neutralization assays were performed in triplicates for each mAb. The table displays only one of the most likely possible D allele assignments.

identical HC VDJ nucleotide sequence of mAb 23 in both BM and spleen. All identified members of this lineage shared the same HCDR3 amino acid sequence, characterized by a single Y106S substitution from the germline derived *IGHJ4-3*. Additional frequently shared SHM positions were a K57T substitution in the HCDR2 and N77T in FR3 (Fig. 5a). We cloned and characterized 7 additional HC members of this lineage identified from the bulk IgG libraries, as well as the germline-reverted HC VDJ sequence, all of which were paired with the mature mAb 23 LC for expression. All the resulting mAbs, including the HC germline-reverted version, potently neutralized D614G. When testing their neutralization breadth against a panel of Omicron subvariants, including BA.1, BA.2, BA.4/5, BA.4.6, BA.2.75, and BA.2.75.2 we found that the germline-reverted mAb 23 neutralized BA.1, BA.2, BA.4/5, and BA.2.75, but not BA.4.6, and BA.2.75.2 (Fig. 5a). Further, we found that breadth extending to BA.4.6 and BA.2.75.2 was acquired already at week 6 (iLN_2315, blood_1089) when the Ab had a very low SHM level (1.7%) (Supplementary Data 4), with similar neutralization breadth displayed by Ab sequences identified in the HC VDJ libraries from week 11 (iLN_3083, blood_14) and week 32 (perLN_80).

Overall, the lineage consisted of sequences from two major clades. Antibodies from the clade shown in the upper part of the tree and alignment (spleen_49, spleen_272 and mAb 23) displayed the broadest activity, with capacity to neutralize all virus variants tested, including improved potency compared to mAb 23 against BA.4.6, BA.2.75 and BA.2.75.2. Antibodies from the lower clade (perLN_298, perLN_67, spleen_177 and spleen_292) displayed reduced breadth, with no activity against BA.2.75.2 and a significant reduction in potency against BA.1, BA.4.6 and BA.2.75. There was no difference in average SHM between sequences belonging to the upper clade, spleen_49, spleen_272 and mAb 23 having 5.4%, 4.8% and 5.4%, respectively, and lower clade, perLN_298, perLN_67, spleen_177 and spleen_292 having 4.8%, 5.4%, 4.3% and 3.7%, respectively (Supplementary Data 4).

To further understand the role of SHM, we investigated the interaction between mAb 23 and a D614G-derived prefusion-stabilized spike by cryo-EM (Fig. 5b and Supplementary Fig. 5). The solved structure revealed the capacity of mAb 23 to bind with the RBD in both "up" and "down" conformations by interacting with a lateral angle with an externally exposed epitope (Fig. 5b, c and Supplementary Fig. 6A, B). Binding was primarily mediated by the HCDR3 residues D99, Y103 and D104. Residue Y53 from the HCDR2 and adjacent LC residues Y49 and A50 also contributed to the binding (Fig. 5d, e and Supplementary Fig. 6C, D). We mapped the epitope to 7 amino acid residues on the RBD surface (T345, R346, N440, K444, N450, Y451, and S494). RBD residues interacting with HCDR3 (R346, K444, N450 and Y451) or with the *IGKV1-NL_15* LC (T345 and N440) were outside of the hACE2 binding footprint, while S494, the only residue within the hACE2 footprint, interacted with the HCDR2 (Fig. 5d, e). Neutralization appears to occur through steric hindrance via the minimal overlap between the epitope and the interaction between RBD and the ACE2 receptor. The mAb 23 epitope is composed of a set of semi-conserved residues[26] in the Omicron panel tested in this study with two notable exceptions: N440K variations were present in all Omicron subvariants, while the R346T mutation was found only in BA.4.6 and BA.2.75.2 (Supplementary Data 5). Interestingly, N440K did not impact cross-neutralization against Omicron, while R346T likely contributes to the loss in potency against BA.2.75.2 (Supplementary Fig. 6E and Supplementary Data 5). Taken together, these data indicate that: (i) mAb 23 lineage selection relied heavily on HCDR3-mediated recognition of the

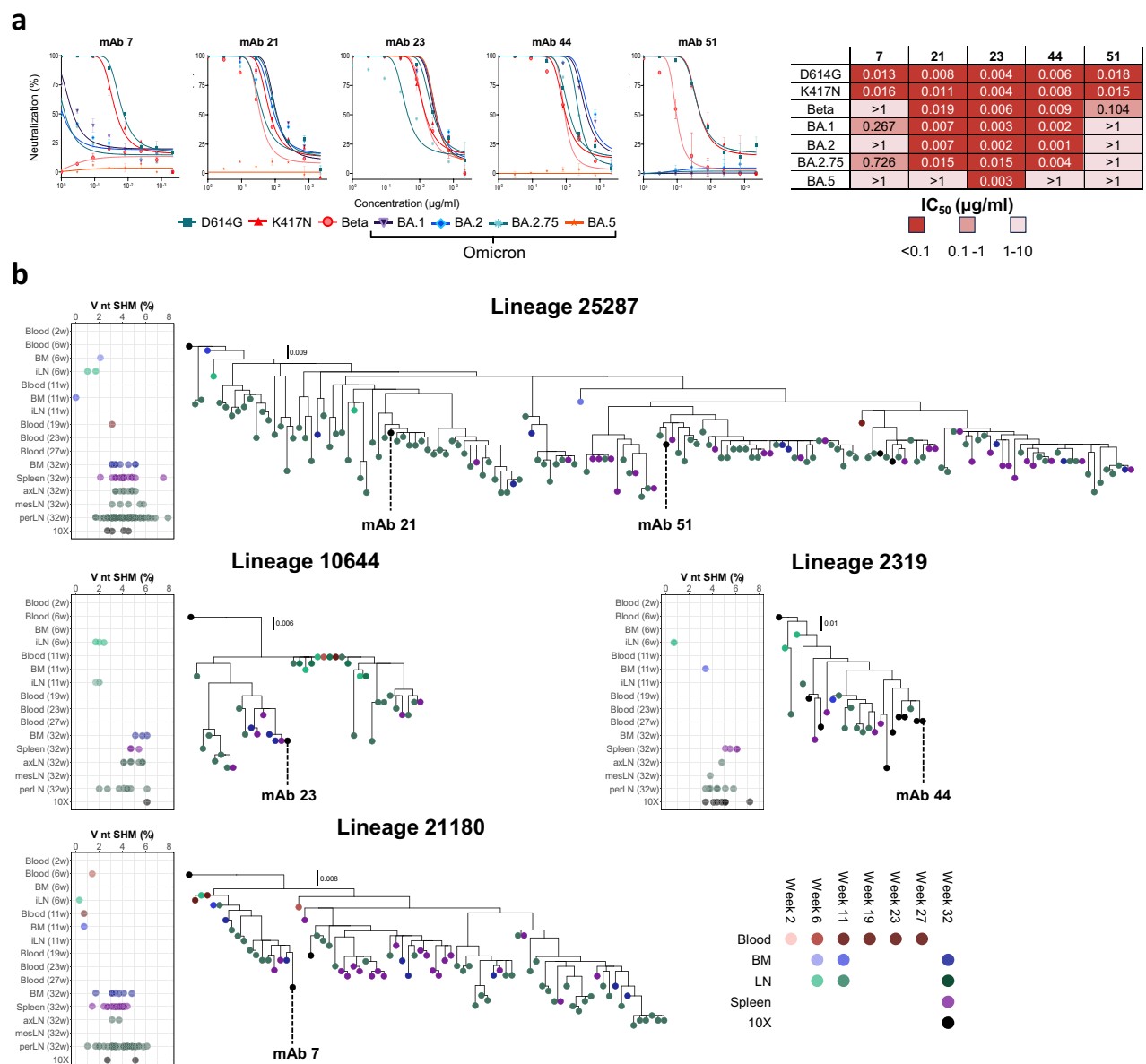

**Fig. 4 | Neutralization breadth and phylogenic analysis of neutralizing Ab lineages. a** Neutralization curves (left) and IC50 values (right) against D614, K417N, Beta and Omicron subvariants BA.1, BA.2, BA.2.75, and BA.5 for the five broadest mAbs. For each curve, the plot displays single data as dots alongside mean values and standard deviation from triplicates. Source data are provided as a Source Data file. **b** Phylogenetic trees of the broadly neutralizing mAbs with the origin of the

traced members displayed in red, blue, green, and purple for blood, BM, LN, and spleen, respectively, with darker shades for increasing weeks. Lineage members obtained from the 10x single-cell paired V(D)J sequencing are displayed in black. A summary of V gene SHM levels at the nucleotide level for each lineage member is displayed to the right of each tree with the compartment of origin and time point of detection shown in temporal order using the same color scheme.

RBD, (ii) changes in neutralization breadth did not rely on the HCDR3 or LC residues, (iii) neutralization breadth can be acquired idiosyncratically in some clades, and (iv) neutralization breadth can be acquired with low SHM early after boosting and persists.

## Discussion

The NHP model is valuable for characterizing immune response dynamics following vaccination, including those for Covid-19[27–41]. With recent improvements in our understanding of macaque IG germline gene variation[42], the model is also amenable to detailed studies of immune repertoires and the isolation of vaccine-induced mAbs[43–45]. In the current study, we characterized expressed B cell repertoires in two SARS-CoV-2 spike protein-immunized macaques by collecting samples from multiple tissues at longitudinal time points, providing detailed information about the evolution of the response. Spike-sorted B cells

were composed of a broad set of HC and LC genes, typical of polyclonal Ab responses[16,46–49], and increasing SHM of queried lineages was evident following sequential boosting.

Our experimental setup enabled comprehensive tracing of antibody sequences encoded by spike-sorted B cells isolated at the termination time point, more than 7 months after the first immunization. The analysis demonstrates that many of the lineages were traceable at all interrogated time points, demonstrating early elicitation followed by extensive clonal diversification of these lineages. B cell lineage persistence was also previously reported from SARS-CoV-2-infected individuals based on the analysis of mAbs or single B cell sequencing data from sequential time points[50–52]. A limitation of these approaches is the number of cells and compartments that can be sampled. Therefore, for a more in-depth examination of Ab lineage evolution following spike immunization, we combined mAb isolation with deep

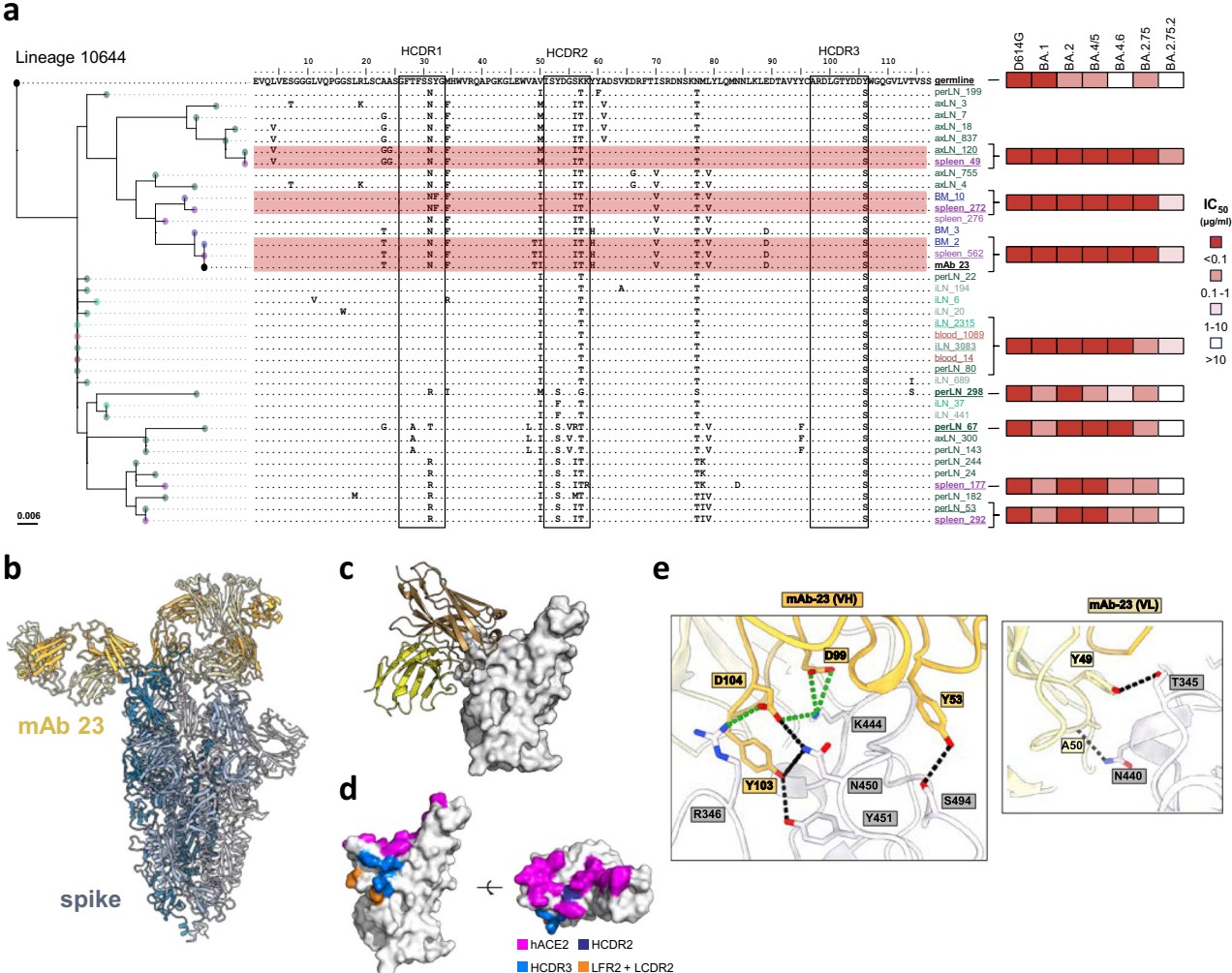

**Fig. 5 | Phylogenetic and cryo-EM analysis of mAb 23 in complex with spike.**
**a** mAb 23 phylogenetic tree, with amino acid alignment of each member. The tree is rooted to its germline, each member is color coded in red, blue, green, and purple for blood, BM, LN and spleen, respectively, with darker shades for increasing weeks. Positions of SHM are indicated and the HCDRs are highlighted by black squares. IC$_{50}$ neutralizing titers against D614G, BA.1, BA.2, BA.5, BA.4.6, BA.2.75, and BA.2.75.2 for expressed members of the lineage are shown to the right in a red color scale. Neutralization assays were performed in triplicates for each member. Source data are provided as a Source Data file. **b** Cryo-EM structure of the mAb 23 Fab interacting with the D614G prefusion spike. **c** Closer-up representation of mAb 23 interacting with the RBD. **d** Contact residues mapped on the RBD. hACE2 is colored purple, HCDR2 in dark blue, HCDR3 in light blue and LC in orange. **e** mAb 23 and RBD residues interaction. mAb23 is colored in yellow while the RBD residues are shown in white/gray. Ionic bonds are represented as dotted lines in green and hydrogen bonds in black.

BCR repertoire sequencing of libraries generated from multiple immune sites and time points, allowing millions of full-length Ab VDJ sequence reads to be queried to trace Ab lineage evolution.

We characterized B cell repertoires in two animals for which we previously reported their peripheral plasma neutralizing response following spike protein vaccination[23]. We focused on one animal, H03, in which we identified as many as 191 spike-sorted B cell lineages, which we could trace in samples from the spleen and non-draining LN, including axLN, mesLN and perLN, indicating widespread dissemination throughout the lymphatics. Tracing of vaccine-induced B cell lineages to distal non-draining LN, such as axLNs, can be attributed to circulation of clones. The perLN was a particularly rich source of vaccine-induced lineages. Previous research has highlighted the presence of vaccine-elicited memory B cells in the perLN, exceeding those found in draining lymph nodes[53]. Additionally, studies involving mRNA antigen tracking after quadriceps immunization in macaques illustrated a distinct pathway for antigen dissemination where the perLN were centrally involved[54]. Taken together, these data suggest that the perLN is an important compartment for B cell differentiation/

trafficking following quadriceps-located immunizations. This argues for the inclusion of non-routinely sampled compartments, like the perLN or other immune cell-rich tissues, as well as B cell lineage tracing in preclinical immunization studies aiming to understand the polyclonal response.

Humoral immunological memory depends in part on the generation of antigen-specific long-lived plasma cells. A substantial portion of these cells are known to reside in the BM, contributing to their longevity. A recent longitudinal study in NHPs showed that adjuvanted SARS-CoV-2 spike protein vaccination established spike-specific long-lived plasma cell reservoirs[22]. In our study, we detected spike-sorted lineages in all longitudinally collected BM samples, including sequences that were identical or clonally related to the broadly neutralizing mAb 23. Previous research suggests that clonotype persistence in BM plasma cell repertoire establishes stable long-term plasma cell populations[55]. Furthermore, recent studies in mice suggest that BM plasma cell survival niches depend on the duration of the response[56,57]. In our study, BM samples were collected from two sources: early samples from humerus aspirates and termination samples from femur

washes. Analysis of IgG repertoires from these samples showed that sequences from the same B cell lineage were present in both compartments. Whether this results from the initial seeding of multiple BM compartments or recirculation of plasma cells between compartments, as recently shown in another study[58], remains unknown.

A question of interest in the vaccine field is if B cells elicited by an initial antigen dominate the response following exposure to a similar but heterologous antigen, for example after vaccination with updated vaccines. Such immune imprinting may limit de novo responses to the new antigen and reduce the ability to stimulate protective immunity against an evolving pathogen. Recent studies have shown that an initial infection or vaccination stimulates cross-reactive B cells that can be boosted by break-through infection with later emerging SARS-CoV-2 variants and after immunization with updated vaccines[15,59–62]. We and others have shown that Ab affinity maturation can improve neutralization breadth in humans infected with SARS-CoV-2[13,14,63]. In the current study, the phylogenetic analysis demonstrated that acquisition of breadth through B cell diversification can occur without compromising neutralization against the original variant used for immunization, in this study D614G. For lineage 10644, of which the broadly neutralizing mAb 23 is a member, we identified two main clades of sequences; one where the neutralization breadth was improved, and another where the neutralization became more specific to the eliciting antigen, the D614G spike. Examining the impact of SHM, individually or in combination, on antibody recognition may further elucidate the evolution of the lineage. Diversification of clonal Ab lineages resulting in clades of antibodies that evolve different neutralization breadth was also previously reported in longitudinal studies of HIV-1 B cell responses[17]. These observations align with the concept that B cell diversification through SHM can prepare the immune system against future variants without compromising the recognition of the original antigen, as also proposed by Longo and Lipsky[64] and recently reemphasized by work from Korenkov et al.[14].

In conclusion, we comprehensively characterized the evolution of B cell lineages induced by SARS-CoV-2 spike immunization across multiple immune compartments. We identified the perLN as a particularly rich source of Ab sequences to study B cell lineage diversification and we showed that antibody maturation modulated neutralization breadth. Thus, multi-compartment longitudinal deep antibody lineage tracing is a powerful approach to study B cell evolution to inform the development of effective vaccine strategies.

## Study limitations
The analysis performed in this study is limited to a small number of immunized macaques ($n = 2$). Additionally, the vaccination regimen (interval between immunizations, dose, format, and injection route) differs from regimens typically used human. A further limitation is the lack of neutralization data on currently circulating variants, which require rigorous experiments that are outside the scope of this study. Final limitations of this study include the use heavy chain only bulk NGS libraries for lineage tracing and the availability of biological samples to allow uniform samplings of the macaques included in the study.

## Methods
### Ethics statement
The animal work was conducted with the approval of the regional Ethical Committee on Animal Experiments (Stockholms Norra Djurförsöksetiska Nämnd). All animal procedures were performed according to approved guidelines, following permits #18427-2019 and #10895-2020.

### Animals' immunization and sample collection
One male (I10) and one female (H03) rhesus macaque (*Macaca mulatta*) of Chinese origin, 4-5 years old, were housed at the Astrid

Fagraeus Laboratory at Karolinska Institutet. Housing and care procedures complied with the provisions and general guidelines of the Swedish Board of Agriculture. The facility has been assigned an Animal Welfare Assurance number by the Office of Laboratory Animal Welfare (OLAW) at the National Institutes of Health (NIH). The macaques were housed in groups in 14 m³ enriched cages. They were habituated to the housing conditions for more than six weeks before the start of the experiment and subjected to positive reinforcement training to reduce the stress associated with experimental procedures. The macaques were weighed at each sampling. All animals were confirmed negative for simian immunodeficiency virus (SIV), simian T cell lymphotropic virus, simian retrovirus type D and simian herpes B virus. All immunizations and samplings were performed under sedation with 10-15 mg/kg ketamine (Ketaminol 100 mg/ml, Intervet, Sweden) administered intramuscularly (i.m.). For macaque immunizations, stabilized spike trimer (100 µg) was mixed in 75 µg of Matrix-M (Novavax AB). Macaques were immunized intramuscularly (i.m.) with half of the dose administered in each quadricep at weeks 0, 4, and 9 and weeks 30 (only I10) and 31 (only H03). Blood samples were collected pre-immunization and at weeks 2, 6, 11, 19, 23, 27, 32 (only I10), and 46 (only I10). Draining iLNs were collected at weeks 6 (right side), 11 (left side), 32 (only H03, at termination), and 46 (only I10, at termination). BM samples were collected at weeks 6 and 11 by aspiration from the humerus and at weeks 32 (only H03, at termination) and week 46 (only I10, at termination) by collection and washing of the femur. Spl samples were collected at weeks 32 (only H03, at termination) and week 46 (only I10, at termination). perLN, axLN and mesLN samples were collected at week 32 (only H03, at termination).

### Recombinant spike and RBD
Recombinant RBD (encoding a C-terminal His-tag) were synthesized (Integrated DNA Technologies) and cloned into a mammalian expression vector (pcDNA3.1), using a Gibson Assembly Mastermix (New England Biolabs). Spike ectodomain was produced in the format of prefusion stabilized with 2 prolines for immunization as reported in Mandolesi et al. [23] and prefusion stabilized with 6 prolines (HexaPro) for ELISAs and Cryo-EM structure resolution as reported in Sheward et al. [65]. Spike and RBD were produced by the transient transfection of Freestyle 293-F cells (ThermoFisher, Cat# R79007) using FreeStyle MAX reagent (Thermo Fisher) and polyethylenimine (PEI), respectively. The HIS-tagged Spike ectodomain and RBD were purified from filtered supernatant using nickel IMAC resin (HisPur Ni-NTA, Thermo Fisher Scientific) followed by size-exclusion chromatography on a Superdex 200 (Cytiva) in PBS.

### Biotinylated spike probe construction
The spike probe was produced by conjugating biotinylated HexaPro spike to streptavidin-allophycocyanin (SA-APC) (Invitrogen). The biotinylated spike was produced using NHS-Chromalink-Biotin (Solulink) with an average ratio of 3:1 (biotin:spike). The labeled spike was purified on PD-10 desalting columns. For the conjugation, 10 µg of spike protein was incubated with 1 µl of SA-APC (1 mg/ml) for 20 minutes on a shaker at 4 °C. The process was repeated until a total of 5 µl SA-APC had been added.

### Single-cell sorting and sequencing
Single cells were sorted with a four-laser FACSAria cell sorter (BD Bioscience) by gating Aqua Live/Dead stain (Life Technologies)–negative, CD3⁻(FITC, BD Pharmingen™, Clone SP34, Cat# 556611), CD14⁻(BV786, BD Horizon™, Clone M5E2, Cat# 563698), CD20⁺(BV421, BD Horizon™, Clone 2H7, Cat# 562873), spike⁺ cells. Spike probes were produced by conjugating a fluorophore via biotinylation as described above. Prior to sorting, TotalSeq-C hashing barcodes for downstream sample identification were added to the following samples collected at week 32: one draining iLN-R1, another

distinct draining iLN-R2, spleen and pooled non-draining LNs, LNOther (mesLN, axLN and perLN) obtained from H03. PBMCs from blood samples obtained from I10 were also labeled with a unique TotalSeq-C barcode. The sorted cells were processed with the 10X Chromium, sequencing '5' V(D)J' enriched libraries, and TotalSeq-C feature barcode libraries. Rhesus Macaques are not a species directly supported by 10X, so we adapted the Human 10X kit by spiking-in a mix of IG constant-region primers from Brochu et al. [66] during the two enrichment PCR steps (Supplementary Table 1). Briefly, 5 µl of each primer (100 µM) were added to MasterMix1 and MasterMix2, for enrichment steps 1 and 2, and diluted to a final volume of 50 µl. Each enrichment steps were spiked with 5 µl of MasterMix after removal of 5 µl nuclease-free water. VDJ read assembly was run in CellRanger (v3.1.0) de novo mode, to avoid reference bias, since CellRanger lacks a Rhesus Macaque reference database. The filtered contigs files were then assigned to individualized H03 or I10 IGH, IGK, and IGL database. Non-productive sequences or sequences without CDR3 identification were removed. Subsequently, we filtered cells with multiple IGH, IGK or IGL assignments and cells with presence of both IGK and IGL. Finally, we set a strict threshold for hashing barcode assignment. The sequences with maximum value of counts per hash lower than 20 were classified as Low_counts. Hashes accounting for >= 60% of all counts were selected for assignment. Otherwise, the sequences were classified as Unassigned.

## HC NGS Library preparation

IgG, multiplexed IgG and IgA, and IgM libraries were prepared following the 5'MTPX protocol described in Vázquez et al. [67] and Vázquez et al. [42]. Briefly, cDNA synthesis was performed using the Sensiscript RT kit (QIAGEN), 200 ng of total RNA and isotype-specific primer containing 21 nt of semi-structured UMI and the Illumina Read2 sequence for 1 h at 37 °C. The cDNA synthesis product was purified (GeneJET PCR purification Kit, Thermo Scientific). The amplification of the second cDNA strand was performed separately with a mix of 5' forward primers targeting the leader sequence of IGHV genes, reported in Vázquez et al. [42], and Read2U using KAPA HiFi Hotstart ReadyMix system (Kapa Biosystems). Indices were introduced in a 10 cycle PCR reaction. The libraries were sequenced with 15% PhiX174 DNA and the Illumina Version 3 (2 × 300 bp) sequencing kit using a MiSeq 2×300 platform (Illumina).

## Individual germline database generation

Individualized germline databased were obtained from IgM, IgK and IgL libraries synthesized from blood samples obtained before the vaccination regimen were used to generate individual germline repertoire via inference analysis performed with the IgDiscover[24] pipeline version v1.0.0 (https://gitlab.com/gkhlab/igdiscover22) with default settings. The reference database for IGH genes was obtained from KIMDB[42] (http://kimdb.gkhlab.se/) by pooling both rhesus and cynomolgus databases and removing duplicates. The reference databases for IGK and IGL genes were obtained from KIMDB by pooling both rhesus and cynomolgus databases and removing duplicates.

## Single-cell paired V(D)J sequences animal and compartment assignment

V(D)J sequences from single-cell analysis were initially assigned to a compartment via TotalSeq-C hashing barcode count. With counts equal to or over 20, a threshold of 60% was used to determine reliable assignment to one compartment. The rest of the sequences were divided into two groups, low_count and unassigned, based on TotalSeq-C count lower than 20 or failure to pass the 60% threshold, respectively. To determine whether these sequences belonged to H03 or I10, they were assigned to each animal's individualized databases and underwent lineage tracing analysis. Only sequences traced to lineages within only one animal were retained and used for downstream analysis.

## Lineage tracing analysis

For each animal, paired V(D)J sequences obtained from single-cell analysis were processed using the IgBlast module from IgDiscover for the assignment of individualized V(D)J germline repertoire for HC and LC. Bulk IgG and multiplexed IgG and IgA libraries were initially processed with IgDiscover for sequence assignment to individualized VDJ germline repertoire and selection of IgG reads. We then performed a denoising step to remove sequencing errors using the Fast Amplicon Denoising (FAD) tool described in Kumar et al. [68] and removed chimeric sequences (likely due to PCR recombination between unrelated antibodies) using a hidden Markov model designed for this purpose (https://github.com/MurrellGroup/CHMMera/), processed reads numbers are in Supplementary Table 2. Lineages were identified using the IgDiscover clonotypes module by combining HC sequences from a single cell and bulk sequencing. Lineages were defined by identical V and J allele assignments, identical CDR3 lengths, and placement into the same single-linkage cluster of HCDR3s with a 0.8 nucleotide identity cutoff. If single-cell sequences with different LC assignments were assigned to the same lineage, the lineage was redefined by also considering the same V and J allele assignment of the LC and the same CDR3 length. In case of divergent LCs, lineage assignment was determined by minimum Levenshtein distance to a single-cell HC sequence as reference. Sequences from each lineage were then aligned using MAFFT v7.490[69]. The alignment was used as input to FastTree[70] (compiled with the double-precision flag) to compute maximum-likelihood phylogenetic trees. The function phylo::reroot from R package phytools v1.2-0 was used to root the tree to its germline sequence. The ancestral HCDR3 germline sequence, used for visualization, was inferred using the package MolecularEvolution.jl (https://github.com/MurrellGroup/MolecularEvolution.jl).

## Monoclonal antibody generation

VDJ sequences from the single-cell analysis were re-adapted for Gibson assembly cloning by inserting complementary overlapping sequences to the expression vectors human IgHγ1 (80795), Igκ1 (80796) or Igλ2 (99575) leader and constant regions[71]. The reaction was carried out by mixing 50 ng of digested vector, 30 ng of VDJ insert and 10 µl of 2X Gibson Assembly Master Mix (New England BioLabs) in a 20 µl reaction and incubating for 1 h at 50 °C. The reaction product was transformed into XL10-Gold Ultracompetent cells (Agilent) and cultures were scaled up to obtain a suitable quantity of plasmid for expression in FreeStyle™ 293 F cells (ThermoFisher, Cat# R79007). Before transfection, the presence of the insert and in the correct reading frame was verified by Sanger sequencing. Transfection was carried out by adding a transfection mix to 1.2–1.5 × 10^6/ml HEK293F cells. Mix for a 30 ml reaction was prepared by mixing 18 ng of HC vector, 18 ng of LC vector and 50 µl of FreeStyle™ MAX Reagent and Opti-MEM in a final volume of 600 µl. The supernatant was harvested after 7 days and mAbs were purified using gravity driven column purification with Protein G Sepharose (Cytivia). From a volume of 30 ml of supernatant, 200 µl protein G Sepharose were loaded in a column and supernatant was applied to the column 5–6 times. After washing with PBS, elution was carried out with 2 ml of elution buffer (0.1 M Glycine.HCl, pH 2.7) into 300 µl of neutralization buffer (1 M Tris-HCl, pH 9). The solution was then diluted in PBS and concentrated using Pierce™ Protein Concentrators with a cutoff of 30 kDa.

## mAb ELISA

ELISA plates were coated with 100 µL of prefusion-stabilized spike or RBD protein at a concentration of 1 µg/ml and blocked for 1 hour at room temperature with 200 µL blocking solution containing 5%(w/v) non-fat milk powder in 1x PBS. MAbs were serially diluted in blocking solution starting from 5 µg/ml were added and incubated for 2 hours at room temperature. Plates were washed 6 times with PBS-T and

antibody-antigen interaction was detected using 100 μL with HRP-conjugated anti-human Fcγ Ab (Southern Biotech, Cat#2015-05)) diluted to 1:10 000 in PBS-T. Plates were washed 6 times with PBS-T, developed using 100 μL of 3,3',5,5″-tetramethylbenzidine (TMB) substrate solution (Invitrogen) per well and stopped using 100 μL of 1 M sulfuric acid per well. OD was read at 450 nm in an Asys Expert 96 plate reader (Biochrom). $EC_{50}$ titers were calculated from the posterior median value midway between the plate minimum and maximum. All experiments were performed in triplicates.

### Pseudovirus neutralization assay

Pseudovirus neutralization assays were performed as described previously[8,9]. Briefly, pseudoviruses sufficient to produce ~100,000 relative light units (RLUs) were incubated with serial three-fold dilutions for 60 min at 37 °C in a black-walled 96-well plate. 10 000 HEK293T-ACE2 cells (in-house) were then added to each well, and plates were incubated for 48 h. Luminescence was measured using Bright-Glo Luciferase Assay System (Promega, Madison, WI, USA) on a GloMax Navigator Microplate Luminometer (Promega). All fold-changes reported use titers from neutralization assays run in parallel. $IC_{50}$ titers were interpolated as the mAb concentration at which RLUs were reduced by 50% relative to the mean of 8 control wells in the absence of mAb. All experiments were performed in triplicates.

### Preparation of Fab fragments

Fab fragments of mAb 23 were prepared by digesting IgG with immobilized papain and separation of Fab and Fc fragments with a Protein A column, using the Pierce Fab Preparation Kit (ThermoFisher Scientific) per the manufacturer's instructions.

### Cryo-EM sample preparation and imaging

Spike trimer (1.2 mg/ml) and a Fab preparation of mAb 23 were mixed in a 1:4 molar ratio (S trimer: Fab), followed by incubation on ice for 10 min. Prior to cryo-EM grid preparation, grids were glow-discharged with 25 mA for 2 min using an EMS 100X (Electron Microscopy Sciences) glow-discharge unit. The grids used were CryoMatrix holey grids with amorphous alloy film (Nitinol; R 2/1 geometry; Zhenjiang Lehua Technology Co., Ltd). 3.5 -μl of sample were applied to the grids and the grids with sample were vitrified in a Vitrobot Mk IV (Thermo Fisher Scientific) at 4 °C and 100% humidity [blot 6 s, blot force 4, 595 filter paper (Ted Pella Inc.)].

Cryo-EM data collection was performed with EPU 2.13 (Thermo Fisher Scientific) using a Krios G3i transmission-electron microscope (Thermo Fisher Scientific) operated at 300 kV in the Karolinska Institute 3D-EM facility. Movies were acquired in nanoprobe EFTEM SA mode at 165 kx nominal magnification with a slit width of 10 eV using a K3 Bioquantum (operated in CDS mode) for 2 s during which 60 movie frames were collected with a fluency of 0.91 e − /Å[72] per frame (Supplementary Table 3). Motion correction, Fourier cropping (to 1.01 Å per pixel), CTF estimation and particle picking were performed on the fly using Warp[72]. Particles were imported into CryoSPARC 3.3.1[73] for further processing. A total of 4,774 micrographs were selected based on an estimated resolution cut-off of 5 Å and defocus below 4 microns. In total 439,641 particles were extracted from the selected micrographs and the extracted particles were used first for 2D classification and good 2-D classes with high-resolution features were selected for two-classes ab-initio model generation followed by two rounds of heterogeneous refinement (with all picked particles). Local CTF refinements were performed interspersed with global aberration estimation and correction (beam tilt, trefoil, tetrafoil, and anisotropic magnification). The three bound Fabs of mAb 23 had varying orientations relative to the body of the spike and relative to each other. Therefore, 3D-VA and 3D-VA display (cluster mode) was performed within CryoSPARC 3.3.1. Two classes from 3D-VA display were selected

for further processing and from the particles (135,612 particles) belonging to these classes we performed particle subtraction followed by local reconstruction of the volume close to one of the RBD-Fab interfaces. This process significantly enhanced the resolvability of the map and thereby enabled molecular fitting and interpretation of the maps (Supplementary Fig. 5). All particles were processed with C1 symmetry. Please see table Supplementary Table 3 for refinement and validation results.

### Cryo-EM model building and structure refinement

The structure of the ancestral D614G spike protein trimer in 1-up conformation PDB: 7A25[74] was used as a starting model for model building. The mAb 23 model was predicted using AlphaFold[75]. The model was mutated and rebuilt manually to reflect the used spike sequence and structure. Structure refinement and manual model building were performed using COOT v0.9.8.91[76] and PHENIX v1.20[77], respectively, in interspersed cycles with secondary structure, Ramachandran, rotamers and bond geometry restraints. Structure figures were generated with UCSF ChimeraX v1.6[78] and PyMol v.2.5.7[79].

### Reporting summary

Further information on research design is available in the Nature Portfolio Reporting Summary linked to this article.

## Data availability

NGS repertoire sequencing data have been deposited in the European nucleotide archive (ENA) with the following accession numbers: from ERR12544449 to ERR12544478. Single-cell repertoire sequencing data have been deposited in the ENA with the following accession numbers: from OZ032182 to OZ034681. Monoclonal antibody sequences are deposited at GenBank with the following accession numbers: from PP208826 to PP208901. The associated accession numbers, coordinates and structure factors of the cryo-EM data reported in this paper are available from the Protein Data Bank (PDB) and the Electron Microscopy Data Bank (EMDB) under the accession codes PDB: 8Q5Y and 8P5M, and EMDB: EMD-18180, EMD-17451. Source data are provided in this paper.

## Code availability

IgDiscover22 v1.0.0 is available at https://gitlab.com/gkhlab/igdiscover22, Scripts used to generate all results in the paper are available at: https://gitlab.com/gkhlab/Multi-compartmental_diversification_of_neutralizing_antibody_lineages_dissected_in_SARS-CoV-2_spike-immunized_macaques and https://doi.org/10.5281/zenodo.11104687. The HMM code for chimera identification is available at https://github.com/MurrellGroup/CHMMera.

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

## Acknowledgements

We thank Dr. Bengt Eriksson and the personnel at the Astrid Fagraeus laboratory for expert assistance with animal experiments. We thank Novavax AB, Uppsala, Sweden, for the Matrix-M adjuvant. All cryo-EM data were collected at the Karolinska Institutet's 3D-EM facility. Analysis of 10X single-cell raw data was enabled, in part, by Anastasios Glaros and resources provided by the National Academic Infrastructure for Supercomputing in Sweden (NAISS) and the Swedish National Infrastructure for Computing (SNIC) at Uppmax, partially funded by the Swedish Research Council through grant agreements no. 2022-06725 and no. 2018-05973. This project was supported by grants from the European Union's Horizon 2020 research and innovation program under grant agreement no. 101003653 (CoroNAb) to G.M.M., G.B.K.H., and B.M. from the Swedish Research Council to B.M. (2018-02381) and to G.B.K.H. (2017-00968), from SciLifeLab's Pandemic Laboratory Preparedness program to B.M. (VC-2022-0028) and G.B.K.H. (VC-2022-0028), from the Erling Persson Foundation (20210125) to B.M. and G.B.K.H., from the he Swedish Research Council to B.M.H. (2017-6702 and 2018-3808) and the Knut and Alice Wallenberg Foundation to B.M.H. We also thank the Fondation Dormeur (Liechtenstein) for its generous contribution towards equipment. Open access funding was provided by Karolinska Institutet.

## Author contributions

M.M., B.M., and G.B.K.H. designed the study, analyzed the results, and wrote the manuscript. L.H., C.K., S.K., and G.M. provided recombinant proteins. M.M. coordinated the NHP immunizations. M.M., X.C.D., and M.A. collected and processed the samples. B.M. and M.G. processed the single-cell raw data from 10x Chromium. M.M. generated the NGS libraries and performed lineage tracing and phylogenetic analysis. M.M. and M.C. performed the IG genotyping. M.Ch. implemented the FAD denoising method. A.S. and M.Ch. implemented the chimera detection method. M.M., L.d.V., M.D., and S.K. cloned, expressed, and characterized mAbs. D.J.S., Y.Y., and J.F. performed neutralization studies. H.D. and B.M.H. Analyzed and solved the cryo-EM structures. M.M., H.D., G.B.K.H., and B.M. visualized the data. B.M., G.B.K.H., B.M.H., and G.M. acquired funding. All authors revised the manuscript and approved the final version prior to submission.

## Funding

## Competing interests

D.J.S. consults for AstraZeneca AB on matters related to monoclonal antibody therapeutics for Covid-19. The remaining authors declare no competing interests.
