## [Peer Review File · Nature Communications]

REVIEWER COMMENTS

Reviewer #1 (Remarks to the Author):

In the study, Mandelosi et al immunize rhesus macaques with an adjuvanted recombinant SARS CoV-2 spike protein 4 times and collect longitudinal samples over a period of 32-46 weeks from the blood lymph nodes and bone marrow as well as from the spleens at necropsy. They use a combination of targeted VH-VL sequencing of spike reactive B cells and deep NGS sequencing of the antibody heavy chains to trace the dissemination and evolution of B cell lineages. This analysis revealed widespread dissemination and maturation of the B cell responses across multiple lineages. They identify an expanded family of B cell clones that show remarkable breadth and potency against omicron variants despite being immunized with the ancestral SARS-CoV-2 spike. Different mutation patterns within this family correspond with neutralizing breadth against Omicron variants.

The manuscript is another advance by an expert team in B cell biology and sheds light on primate B cell responses to immunization.

Overall the manuscript is technically sound and well written.

I have a few comments that the authors might consider in a revised manuscript.

-The immunizations are experimental and are not be representative of typical vaccine regimens used in humans. For example, the animals were immunized 4 times over a period of 32 weeks. Moreover the vaccine is administered in the quadriceps. Moreover most people received mRNA vaccines before protein based vaccines similar to the one used in this study were developed. It might be worth noting that these differences may not be reflective of the general population as it relates to SARS-CoV-2 immunity.

It's interesting that members of the mab 23 lineage that do not neutralize BA.2.75.2 are mutated a the Y53 residue (S or F) which makes contact with S494 on the spike. Were these mutations selected for enhanced binding to the vaccine spike at a cost of potency against the BA.2.75.2 variant?

Reviewer #2 (Remarks to the Author):

In this report by Mandolesi et al, the authors performed antibody isolation and characterization along with B cell receptor sequencing in two macaques immunized 4 times with SARS-CoV-2 spike

protein. They examined and tracked specific mAb lineages through multiple tissues and identified lineages with varying breadth of neutralization against emerging omicron subvariants, up to BA.5. Key findings include the identification of a single mAb capable of a remarkably broad degree of neutralization against all tested viral isolates, persistence of particular lineages from prime through the end of the study and the identification of the structural basis for neutralization breadth of particular mAbs. The report is well written and the studies very well executed. In addition to the limitations articulated in the report, the small sample size (n=2) is also a limitation. However, the extensive characterization along with longitudinal analyses more than make up for the small number of animals. Beyond this, I have only very minor suggestions.

The abstract does not mention the study used macaques. This minor addition would help orient readers. Obviously a very small issue.

This is beyond the scope of the paper, and not critical for publication, but based on structural data, can the authors speculate on the neutralization breadth of more recent isolates (e.g. XBB, EG.5, JN.1) of mAbs they identified to have broad neutralization? This would enhance clinical relevance since the most recent variant studied is BA.5, which has not been in circulation in quite some time.

Reviewer #2 (Remarks on code availability):

This is outside my area of expertise

Reviewer #3 (Remarks to the Author):

Mandolesi et al. present a comprehensive analysis of B-cell clonal lineage diversification in non-human primates following SARS-CoV-2 vaccination. They employed next-generation sequencing to track this diversification using prefusion-stabilized spike protein adjuvanted with Matrix-M. The study focused on two rhesus macaques, previously characterized for plasma neutralizing response by the same group. Despite the limited sample size, the extensive data generated, along with the labor and expense involved in dataset generation and monoclonal antibody production for downstream testing in pseudoviral neutralization assays, justify publication.

The researchers utilized IgDiscover, a validated analysis program, to analyze repertoire data. They have also made all associated code and scripts for figure generation publicly available on GitHub. Notably, two significant observations regarding vaccine-elicited B cells were made:

Firstly, B-cell lineages continued to evolve and diversify through somatic hypermutation while maintaining specificity and neutralization activity against the ancestral D614G strain, indicating robust immune response without compromise.

Secondly, a striking finding was that a majority (89%) of spike-specific B-cell lineages, identified via antigen-specific flow cytometry, originated from the periaortic lymph nodes. This suggests the periaortic lymph nodes as an active site for B-cell recirculation and a robust source of vaccine-activated B cells. (Moody and colleagues, for example, have made prior observations regarding the occurrence of flow-cytometric spike-reactive B cells in perLN, but Mandolesi has refined and elevated the observation at the molecular level.)

Major Concerns: NONE

Minor Concerns: Although prohibitively expensive, it would have been optimal to run individual 10x scRNA-seqs, rather than TotalSeq-C hashing.

NCOMMS-24-08476A

RESPONSE TO REVIEWER COMMENTS

Reviewers' comments in *italics*, authors' response in blue normal font.

Reviewer #1

In the study, Mandelosi et al immunize rhesus macaques with an adjuvanted recombinant SARS CoV-2 spike protein 4 times and collect longitudinal samples over a period of 32-46 weeks from the blood lymph nodes and bone marrow as well as from the spleens at necropsy. They use a combination of targeted VH-VL sequencing of spike reactive B cells and deep NGS sequencing of the antibody heavy chains to trace the dissemination and evolution of B cell lineages. This analysis revealed widespread dissemination and maturation of the B cell responses across multiple lineages. They identify an expanded family of B cell clones that show remarkable breadth and potency against omicron variants despite being immunized with the ancestral SARS-CoV-2 spike. Different mutation patterns within this family correspond with neutralizing breadth against Omicron variants. The manuscript is another advance by an expert team in B cell biology and sheds light on primate B cell responses to immunization. Overall the manuscript is technically sound and well written.

We thank the Reviewer for their positive comments, and we appreciate the recognition of our work as an important contribution to the field of B cell immunology.

I have a few comments that the authors might consider in a revised manuscript. -The immunizations are experimental and are not be representative of typical vaccine regimens used in humans. For example, the animals were immunized 4 times over a period of 32 weeks. Moreover the vaccine is administered in the quadriceps. Moreover most people received mRNA vaccines before protein based vaccines similar to the one used in this study were developed. It might be worth noting that these differences may not be reflective of the general population as it relates to SARS-CoV-2 immunity.

The Reviewer fairly notes the differences between the immunization regimen used in our study and typical vaccine regimens employed in humans. To address this, we updated the **Study limitation section** accordingly to highlight the difference between our experimental setup and the general population vaccination experience in **line 329-330** of the manuscript.

It's interesting that members of the mab 23 lineage that do not neutralize BA.2.75.2 are mutated at the Y53 residue (S or F) which makes contact with S494 on the spike. Were these mutations selected for enhanced binding to the vaccine spike at a cost of potency against the BA.2.75.2 variant?

Additionally, the Reviewer asks if SHM in position Y53 in the non-broad clade of the mAb 23 lineage were selected for enhanced binding to the spike variant used for immunization at a cost of BA.2.75.2 neutralization.

We agree that this may be the case, but it may also be a neutral change. We updated the **Discussion** section to mention these possibilities in **line 313-315** of the manuscript.

Reviewer #2

In this report by Mandolesi et al, the authors performed antibody isolation and characterization along with B cell receptor sequencing in two macaques immunized 4 times with SARS-CoV-2 spike protein. They examined and tracked specific mAb lineages through multiple tissues and identified lineages with varying breadth of neutralization against emerging omicron subvariants, up to BA.5. Key findings include the identification of a single mAb capable of a remarkably broad degree of neutralization against all tested viral isolates, persistence of particular lineages from prime through the end of the study and the identification of the structural basis for neutralization breadth of particular mAbs. The report is well written, and the studies very well executed. In addition to the limitations articulated in the report, the small sample size (n=2) is also a limitation. However, the extensive characterization along with longitudinal analyses more than make up for the small number of animals.

We kindly thank the Reviewer for their thoughtful and constructive comments and for appreciating our study. The Reviewer rightfully highlights that we had not mentioned the small sample size (n=2) in the Study limitation section. We have now updated the **Study limitations section** to mention this in **line 328-329** of the manuscript.

Beyond this, I have only very minor suggestions. The abstract does not mention the study used macaques. This minor addition would help orient readers. Obviously a very small issue.

We would like to respectfully point out that we did include the species used in our study in the **Abstract (line 26 of the manuscript)**, quoting “Here, we combined monoclonal antibody (mAb) isolation with deep B cell receptor (BCR) repertoire sequencing of **rhesus macaques** immunized with prefusion-stabilized spike glycoprotein”. We believe this may have been overlooked and we agree with the Reviewer that this is important to mention. The title of the manuscript also states that the study was performed in macaques.

This is beyond the scope of the paper, and not critical for publication, but based on structural data, can the authors speculate on the neutralization breadth of more recent isolates (e.g. XBB, EG.5, JN.1) of mAbs they identified to have broad neutralization? This would enhance clinical relevance since the most recent variant studied is BA.5, which has not been in circulation in quite some time.

Finally, the reviewer points out that the SARS-CoV-2 variants included in this study are no longer a clinical concern as they have been replaced by other circulating variants.

Despite structural data being highly informative, we chose to be careful regarding speculations about neutralizing activity of other variants. We know that the epitope recognized by mAb 23 is not completely conserved in all the SARS-CoV-2 variants. For instance, using the examples provided by the Reviewer, the XBB and EG.5 spike glycoproteins contain the same degree of variation as BA.4.6 and BA.2.75.2 including an R346T mutation, while JN.1 carries a N450D mutation. We can speculate that XBB and EG.5 are more likely to be neutralized by mAb23 than JN.1, since we know that R346T is tolerated. However, additional mutations outside of the epitope might impact the tertiary structure of the protein with unpredictable impact on antibody recognition. To address the Reviewer’s question, we updated the **Study limitation section** on **line 330-332** of the manuscript.

Reviewer #3

Mandolesi et al. present a comprehensive analysis of B-cell clonal lineage diversification in non-human primates following SARS-CoV-2 vaccination. They employed next-generation sequencing to track this diversification using prefusion-stabilized spike protein adjuvanted with Matrix-M. The study focused on two rhesus macaques, previously characterized for plasma neutralizing response by the same group. Despite the limited sample size, the extensive data generated, along with the labor and expense involved in dataset generation and monoclonal antibody production for downstream testing in pseudoviral neutralization assays, justify publication. The researchers utilized IgDiscover, a validated analysis program, to analyze repertoire data. They have also made all associated code and scripts for figure generation publicly available on GitHub.

Notably, two significant observations regarding vaccine-elicited B cells were made: Firstly, B-cell lineages continued to evolve and diversify through somatic hypermutation while maintaining specificity and neutralization activity against the ancestral D614G strain, indicating robust immune response without compromise. Secondly, a striking finding was that a majority (89%) of spike-specific B-cell lineages, identified via antigen-specific flow cytometry, originated from the periaortic lymph nodes. This suggests the periaortic lymph nodes as an active site for B-cell recirculation and a robust source of vaccine-activated B cells. (Moody and colleagues, for example, have made prior observations regarding the occurrence of flow-cytometric spike-reactive B cells in perLN, but Mandolesi has refined and elevated the observation at the molecular level.)

Major Concerns: NONE

We would like to kindly thank the Reviewer for their thoughtful and constructive comments, and for generously summarizing some of the key points of our study.

Minor Concerns: Although prohibitively expensive, it would have been optimal to run individual 10x scRNA-seqs, rather than TotalSeq-C hashing.

The Reviewer suggests that running individual 10x sc-RNA seq for each sample would have been advantageous compared to the TotalSeq-C hashing used in our design. We agree that this would have resulted in a more comprehensive 10x sc dataset; however, as also noted by the Reviewer, this would have been very costly. We believe that the combination of 10x sc-RNA seq and bulk IG repertoire analysis by NGS as used in our study provided a rich dataset for lineage tracing, which was suitable for the questions we wished to address in this study.

Reviewers' Comments:

Reviewer #4:

Remarks to the Author:

The cryo-electron microscopy is very carefully performed with clear details of the procedure presented in the text and in the Fig S5.

Minor points

- Line 228: I suggest "...interacting at a lateral angle with..."
- Line 516: please describe the procedure for obtaining the Fabs.
- S6 – The authors could use the whole page for showing better the details from the individual panels. For instance, the fit in panel c is not visible even when zooming in.

NCOMMS-24-08476B

RESPONSE TO REVIEWER COMMENTS

Reviewers' comments in italics, authors' response in blue normal font.

Reviewer #4

The cryo-electron microscopy is very carefully performed with clear details of the procedure presented in the text and in the Fig S5.

We sincerely thank the reviewer for their thoughtful evaluation of our structural work and for the constructive suggestions to our manuscript.

Minor points

- Line 228: I suggest "...interacting at a lateral angle with..."

The reviewer raise three minor points. The first suggest a minor revision of the main text in the **Result** section (line 228 of the manuscript) which we promptly updated.

- Line 516: please describe the procedure for obtaining the Fabs.

Additionally, the reviewer justly points out that the procedure to produce Fabs used for resolving the structure is missing in the **Methods** section. To address this, we have updated the Methods section to include a new subheading titled "Preparation of Fab fragments" in **lines 515-518** of the manuscript.

- S6 – The authors could use the whole page for showing better the details from the individual panels. For instance, the fit in panel c is not visible even when zooming in.

Finally, the reviewer justly highlight insufficient visualization capacity in Supplementary Figure 6. To fix this, we have increased the size, repositioned, and improved the resolution of the panels in **Supplementary Figure 6** to ensure all relevant details are clearly visible.

Once again, we thank the reviewer for their valuable feedback and comments which improved the clarity and quality of our manuscript.